# MONET: The Minor Body Generator Tool at DART Lab

**DOI:** 10.3390/s24113658

**Published:** 2024-06-05

**Authors:** Carmine Buonagura, Mattia Pugliatti, Francesco Topputo

**Affiliations:** Department of Aerospace Science and Technology, Politecnico di Milano, 20156 Milan, Italy; mattia.pugliatti@polimi.it (M.P.); francesco.topputo@polimi.it (F.T.)

**Keywords:** asteroid, small body, procedural generation, morphing

## Abstract

Minor bodies exhibit considerable variability in shape and surface morphology, posing challenges for spacecraft operations, which are further compounded by highly non-linear dynamics and limited communication windows with Earth. Additionally, uncertainties persist in the shape and surface morphology of minor bodies due to errors in ground-based estimation techniques. The growing need for autonomy underscores the importance of robust image processing and visual-based navigation methods. To address this demand, it is essential to conduct tests on a variety of body shapes and with different surface morphological features. This work introduces the procedural Minor bOdy geNErator Tool (MONET), implemented using an open-source 3D computer graphics software. The starting point of MONET is the three-dimensional mesh of a generic minor body, which is procedurally modified by introducing craters, boulders, and surface roughness, resulting in a photorealistic model. MONET offers the flexibility to generate a diverse range of shapes and surface morphological features, aiding in the recreation of various minor bodies. Users can fine-tune relevant parameters to create the desired conditions based on the specific application requirements. The tool offers the capability to generate two default families of models: rubble-pile, characterized by numerous different-sized boulders, and comet-like, reflecting the typical morphology of comets. MONET serves as a valuable resource for researchers and engineers involved in minor body exploration missions and related projects, providing insights into the adaptability and effectiveness of guidance and navigation techniques across a wide range of morphological scenarios.

## 1. Introduction

Minor bodies exploration is an expanding field. These bodies provide incomparable information on the solar system formation and evolution, pose a constant threat to Earth, and serve as a potential source of precious materials that could be exploited [1]. Up until now, several missions such as Osiris-Rex [2], Rosetta [3], Hayabusa 1 [4], and 2 [5], Lucy [6], and DART [7] have been launched towards these targets, while others, such as Hera [7] with its two CubeSats, Milani [8] and Juventas [9], and M-ARGO [10], are planned for the future.

Image-processing (IP) algorithms necessarily need images acquired via an imaging sensor in order to generate optical observables. However, their performance strongly depends on the shape and surface morphological features of the small body, such as craters and boulders. This implies that such methods need to be robust in the face of these types of uncertainties. This work focuses on the procedural generation of small-body characteristics, which stimulate the virtual imaging sensor and allow for the design and testing of robust IP algorithms. The uncertainty arises because minor celestial bodies showcase a variety of traits, encompassing a broad spectrum of shapes, sizes, compositions, and morphological features [11]. The shapes can range from spherical to elongated, irregular, and bilobed, with sizes spanning from a few meters to thousands of kilometers [12]. These bodies exhibit diverse structures, from rubble-pile formations, where boulders are bound together by mutual gravity [13], to monolithic blocks, often assumed for smaller and fast rotators. All these characteristics result in the difficulty to estimate from ground-based observations. Examples include past missions such as DART [14], Osiris-Rex, and Rosetta, highlighting uncertainties surrounding the main physical parameters of celestial bodies, specifically the (65803) Didymos binary system, (101955) Bennu, and 67P/Churyumov-Gerasimenko. Initially, Didymos, the primary body of the (65803) Didymos binary system, was expected to be a diamond-shaped asteroid with an equatorial bulge [15]. However, the DART approach revealed its significantly oblate shape [16]. Similarly, a different surface morphology than expected was observed for (101955) Bennu [17]. The denser distribution of boulders on the surface required a redevelopment of a major subsystem of the mission to guarantee a successful sampling of the surface. Additionally, comet 67P/Churyumov–Gerasimenko was expected to have a diamond shape from ground-based observations. It was only upon Rosetta reaching the comet that its bilobed shape was realized [18].

As mentioned in [19], various techniques are employed to estimate the shape and morphological characteristics of a minor body. The simplest method is light curve analysis, which relies on distant observations of the light variability from the minor body. In practice, photometric observations of a body taken over an extended period are utilized to measure variations in reflected light, providing information about the shape. Many available 3D models are based on this technique (https://astro.troja.mff.cuni.cz/projects/damit/asteroids/browse accessed on 24 March 2024), offering a rough approximation. Alternatively, radar range-doppler imaging, frequently applied to bodies in close proximity to Earth, utilizes radio telescopes. This method allows for higher accuracy in shape and spin estimation compared to the aforementioned technique. Another approach involves high-resolution imagery, leveraging a combination of images taken at different viewing geometries and phase angles. Typically used in rendezvous and flyby missions, this method achieves uniform resolutions. The most accurate method is once again employed in close proximity, and it involves the use of LIght Detection And Ranging (LIDAR). Flyby targets cannot often have global shape reconstruction, due to the limited observation time and constrained geometry.

The shape of the body is a crucial parameter for selecting the most effective Visual Based Navigation (VBN) and Image Processing (IP) method, which must exhibit robustness in the face of estimation uncertainties [20]. Assessing the robustness of processing techniques involves testing them on a wide variety of shapes and surface features, necessitating large image datasets. Moreover, data-driven methods demand large amount of data for training, validation, testing, and generalization purposes [21]. The availability of a variety of shapes and morphological features is fundamental in developing generalized approaches.

Researchers have found it difficult to access advanced tools for procedural modification of minor bodies due to limited availability and accessibility. These tools are often limited or kept confidential for strategic reasons because their development requires a lot of time and expertise, discouraging their implementation. To the best of the author’s knowledge, other tools currently available in the literature encompass both the entire process of image generation from existing minor body shapes, such as SISPO [22], AstroSym [23], and the simulations tools illustrated in [24,25,26]. Alternatively, some tools focus on accurately reproducing surface morphologies, including craters and roughness, by incorporating noise functions, as demonstrated by AstroGen [27,28]. Moreover, various computer graphics software options are available for modeling and rendering minor bodies. High-fidelity rendering softwares like ESA’s PANGU 6.01 [29] and Airbus Defence & Space’s SurRender 7 [30] are constrained by software licenses. In contrast, open-source software such as POV-Ray 3.8.0 (http://www.povray.org/ accessed on 19 January 2024) and Blender 4.1.1 (https://www.blender.org/ accessed on 19 January 2024) are available, although these are not specifically designed for rendering celestial objects. The primary distinction between the latter two lies in the fact that Blender is well-documented, supports Python 3.12 scripting, and has a robust community. This is why Blender has been chosen for this work. However, creating realistic minor bodies manually in Blender can be labor-intensive. To address this, a Python code has been developed to achieve the same results in a more streamlined and procedural manner. The Blender/Python API is exploited for this purpose, indeed, Blender embeds a Python interpreter provided by the “bpy” module that can be imported in a script and gives access to Blender data, classes, and methods. This approach allows to access by script all the Blender functionalities normally accessed via the graphical user interface of the software.

This work presents the Minor bOdy geNErator Tool (MONET), designed by the Deep-space Astrodynamics Research & Technology (https://dart.polimi.it/ accessed on 19 January 2024) (DART) group and used to support the validation and testing of IP and VBN algorithms as well as to construct datasets for artificial intelligence applications [31].

MONET provides a versatile tool for the generation of realistic minor body shapes through the interpolation of existing models. Beyond this, the tool has the capability to produce a diverse array of morphological conditions for minor bodies. These multiple features make it exceptionally powerful, particularly in evaluating the robustness of techniques such as IP and VBN. By allowing the generation of a broad spectrum of minor body shapes, MONET becomes an invaluable resource for comprehensive assessments, providing insights into the adaptability and effectiveness of IP and VBN techniques across a wide range of morphological scenarios. The tool is openly available at https://github.com/MattiaPugliatti/corto (accessed on 24 March 2024) while it is being maintained and further developed.

The paper is structured as follows. Section 2 reports an explanation of the procedural operations considered to modify the original model. In Section 3, results that can be achieved with the tool are presented. Some final considerations are then discussed in Section 4.

## 2. Methodology

This work aims to present a tool capable of generating a wide range of high-fidelity small body models in a procedural manner. This encompasses not only the overall shape of the body but also its surface properties, including roughness, craters, and the distribution and characteristics of boulders. Datasets can be easily built and exploited to validate and test the robustness of IP and VBN algorithms as well as to train, validate, and test data-driven methods. In the following sections, the procedure employed for generating realistic surface morphological features is outlined. The starting point of the tool is a small-body three-dimensional model whose shape can be rough or highly accurate depending on the observation technique employed to recreate it. Therefore, a significant accomplishment of MONET is its capability to procedurally alter surface features, enabling the generation of realistic morphological models of minor bodies. These models are then used as inputs to the Celestial Object Rendering TOol (CORTO) [32]. CORTO takes care of positioning all objects in the scene, including light sources, bodies, and camera positions and orientations. Additionally, it incorporates a reflectivity model and handles the rendering process.

Moreover, there are two default surface morphology features that MONET can recreate. Firstly, a rubble-pile body, such as (101955) Bennu, that presents an elevated number of boulders of different sizes [33]. Secondly, a comet-like surface morphology, with the alternation of rough regions of rocks and smooth sand ones like 67P/Churyumov–Gerasimenko [34], with the presence of small boulders evenly distributed on the body surface.

Blender has been used for the entire procedural generation process. Indeed, various functionalities and operations will be cited in the following sections. To aid the reader’s understanding, brief explanatory footnotes are introduced when topics are not extensively discussed in the text. Further information can be found on the Blender manual website (https://docs.blender.org/manual/en/latest/ accessed on 29 April 2024).

### 2.1. Model Refinement

First of all, a model object is taken from existing databases (https://astro.troja.mff.cuni.cz/projects/damit/asteroids/browse accessed on 20 January 2024), (https://sbn.psi.edu/pds/shape-models/ accessed on 20 January 2024) and imported in Blender. The starting model can be rough or highly accurate depending on the observation technique used to recreate the model. In this step, it is important to ensure that the model does not exhibit mesh artifacts, leading to a non-uniform mesh distribution, as shown in Figure 1. If artifacts are present, *remesh* (It is a tool for generating new mesh tolopology) and *triangulate* (It converts all faces in a mesh to triangular faces) modifiers are applied to correct the mesh. To take advantage of the same settings for all the models imported within the tool and to reduce computational effort, they are resized to the same average size by inscribing them in a 1 m radius sphere. To create a realistic rocky surface, the subsequent step involves smoothing the mesh of the starting model. This task is achieved by utilizing the *subdivision surface* modifier, applied twice. Its function is to split each primitive polygon contained in the mesh, increasing the vertices number of the model giving it a smooth appearance. The Catmull-Clark [35] subdivision method is employed to achieve this objective, creating new vertices based on the averages of the original points. Following the application of these modifiers, the *smooth shading* (Face normals are interpolated to change the way the shading is calculated across the surfaces) functionality is utilized to further enhance the smoothness. Figure 2 shows the model surface improvements after applying the aforementioned modifiers. For models with large number of vertices, like comet 67P/Churyumov–Gerasimenko and asteroid (25143) Itokawa, shown in Figure 3, there is no need to use modifiers, as the existing meshes are sufficiently accurate.

If these modifiers are still applied, the computational workload to execute them might exceed the available computational power. A potential solution to address this challenge and reduce rendering times is to initially employ a *decimate modifier* to decrease the number of faces. However, it has been demonstrated that the saved time is not significant compared to the realism of the final model. The reasons for which this modifier has been discarded are outlined in the following paragraph.

Indeed, this modifier yields models with shapes comparable to highly accurate ones and effectively reduces the rendering time. However, the drawback is a notable increase in the computational time due to the high cost of face reduction. Consequently, for models with an arbitrary number of vertices exceeding 10,000, only *smooth shading* is employed. In the case of models with a low count of faces, like (21) Lutetia in Figure 3, the *decimate modifier* effectively reduces the computational times. In this case, the final model will be smoother with a less realistic distribution of shadows, especially along the terminator, resulting in a segmented line as illustrated in Figure 4.

### 2.2. Morphing

Morphing describes the capability to transition from one shape to another given two starting models. This capability would allow testing IP and VBN algorithms, especially limb-based ones, on a wide variety of shapes, facilitating the interpolation of existing minor bodies and enabling the generation of plausible bodies whose shape can be considered to be part of the envelope of existing minor bodies. Blender embeds this functionality using the *shape keys*. Once two meshes are within the virtual environment, vertices from one mesh are projected onto the shape of the other using the *shrinkwrap* modifier. At this point, it is possible to morph the mesh from its original shape to the projected one. The final shape significantly depends on the number of vertices of the starting mesh, achieving better results when the number of vertices is equal to or higher than the number of vertices of the mesh onto which they are projected. The vertices of the newly generated body are obtained by linearly interpolating the body vertices $V_{b}$ with the projected ones $V_{p}$ according to the following relation:(1)$$V_{m}=wV_{p}+\left(1-w\right)V_{b}$$
where $V_{m}$ represents the vertices matrix of the morphed model, while *w* is the morphing weight, ranging from 0 to 1.

To ensure the morphing strategy generates plausible shapes, shape parameters were employed, namely elongation el, flatness fl, and irregularity ir [36]. They are determined solving a linear least-square problem, fitting an ellipsoid to the point cloud [37]. Once the semi-axes of the ellipsoid (a,b,c) are computed, with a≥b≥c, the elongation and flatness can be calculated as follows: (2)$$\begin{array}{c} el=1-\frac{b}{a} \end{array}$$(3)$$\begin{array}{c} fl=1-\frac{c}{b} \end{array}$$

Regarding irregularity, it is determined by identifying the mesh vertex closest to the body center of mass $r_{b}^{min}$, and projecting this vertex onto the fitting ellipsoid to obtain $r_{b}^{proj}$. The irregularity is then quantified as the ratio between the distances of these two points:(4)$$\begin{array}{c} ir=1-\frac{r_{b}^{min}}{r_{b}^{proj}} \end{array}$$

These metrics range from 0, indicating a perfectly spherical surface, to 1, representing an asymptotic value for extremely elongated, flat, and irregular bodies.

### 2.3. Surface Morphology

Surface morphology is the most important aspect in order to have a model visually resembling a minor body. The previously described shape estimation techniques, apart from imagery and LIDAR, do not allow the reconstruction of the morphological characteristics of minor bodies. Therefore, it is essential to enhance the surface of a simple shape model by introducing morphological features such as roughness, craters, color, and boulders. This is beneficial especially for the validation and testing of feature-based IP and VBN methods.

A material is applied to the object to achieve a rough surface with craters, while boulders are physical three-dimensional objects randomly scattered across the surface.

#### 2.3.1. Surface Roughness

The initial step to achieve realistic minor bodies involves introducing surface roughness. The material is applied to the model mesh using the node tree available in the *shading editor* in Blender. To break the uniformity of color, a *noise* texture is first incorporated as illustrated in Figure 5. Subsequently, the *noise* texture is fed into a *bump* node (it generates a perturbed normal from a height texture), allowing for normal displacement of the object’s surface to recreate the typical roughness found on rocky bodies. The node trees exhibit slight variations for the two default bodies, with the comet-like model characterized by alternating smooth and rough regions. The Blender node trees necessary for accurately reproducing the surface of the bodies are provided in the Appendix A.

#### 2.3.2. Craters

Craters, formed by impact events, are depressions characterized by a roughly circular shape, resulting from the ejection of material in all directions after impact [38]. These features are generated using the *voronoi* texture, which is characterized by multiple circles. To achieve the effect of an excavation, a *color ramp* (it is used for mapping values to colors with the use of a gradient) node is added with four different shades of black. It is then required to combine this texture with the surface roughness one, exploiting the Blender *mix* node (which mixes images working on individual and corresponding pixels). It works on the individual and corresponding pixels of the two input textures. The main problem to solve is given by the protraction of the nearby roughness features inside the crater. Thus, the first operation performed is the subtraction of the craters texture from the surface roughness one. This yields a texture with the same pattern as the surface roughness one but with white circles distributed across it. To convey the impression of an excavation, the craters must be black so that the software moves them towards the inner part of the model. Therefore, a multiplication is performed between the previously generated texture and the craters one. As previously described, the *bump* node is exploited to displace the surface of the object creating a realistic rough surface with craters. The crater generation process is depicted in Figure 6. This approach is executed twice in the node tree using different sizes of *voronoi* textures to generate overlapping craters of varying sizes and locations. Specifically, smaller and larger craters are created. The primary distinction is that the texture for larger craters is not combined with the roughness one. Finally, the textures are blended using the *mix* node. The complete node trees for both small and large craters are illustrated in the Appendix A.

#### 2.3.3. Boulders

Boulders are distributed over the surface of a minor body, and their size can range from small to large. Using the *rock generator* (https://github.com/versluis/Rock-Generator accessed on 20 January 2024) extension in Blender, an arbitrary number of boulders, by default 50, are created, and the same material described in Section 2.3.1 is applied to each of them. This limited number is chosen to minimize the computational load, but it proves sufficient, as the rocks are strategically placed on the body surface using a particle system. As depicted in Figure 7, this is a functionality within Blender that enables the duplication and placement of objects on a surface when the “hair” type is selected in the particle system. Users can choose the number of particles or rocks, along with their size, phase angle, and randomness. The last parameter is set to its maximum value to achieve the most random configuration and avoid repetitions. In Figure 7, a configuration with an elevated number of boulders is shown. Similar to the craters, different numbers of particle systems can be generated, resulting in a wide variety of boulder dimensions, positions, and orientations. By default, three different particle systems are set up for small, medium, and large boulders.

In Figure 8, the main building blocks of the Python code are synthetically summarized. Additionally, Figure 9 presents a simplified version of the designed node tree for introducing craters and surface roughness on the minor body surface. Finally, the entire procedural change process, from the rough 3D model to the refined one, is depicted in Figure 10.

## 3. Results

In this section, the tool capacities are showcased by generating various minor bodies with distinct morphological characteristics. Initially, 100 different shape models are rendered, incorporating a random combination of boulders and rocks, surface roughness intensity, poses, surface color, and reflectivity models, as depicted in Figure 11. This mosaic is created to demonstrate the tool capability to generate a diverse range of models. It is evident that MONET achieves realistic rendering irrespective of the body shape, orientation, and illumination. The object, camera, and light positioning in the scene, as well as the implementation of the reflectivity model, are executed through CORTO. Specifically, a 30 deg Sun phase angle is considered for all the bodies. The reflectivity models used, apart from the well-known Blender-based reflectivity model, Principled BSDF, are (1) Lommel-Seeliger [39], (2) ROLO [40], (3) Akimov [41,42], (4) Linear Akimov [43], (5) Lunar Lambert [44], and (6) Minnaert [45].

Figure 12 illustrates four differently shaped bodies, from extremely regular to irregular, namely, an ideal sphere and ellipsoid, (4) Vesta, and 67P/Churyumov–Gerasimenko, all rendered from the same perspective and lighting conditions, featuring four levels of increasing roughness intensity, craters, and the number of boulders in Figure 12b–d, respectively. The values of the morphology features chosen for the above mentioned mosaic are reported in Table 1. It can be observed that the value for small craters is progressively increased to generate a greater number of craters with smaller average sizes. Conversely, large craters are initially absent (value equal to 0), and then their value is decreased from 16 to 3. With this strategy, as we move from left to right, the contrast in size between small and large craters becomes more pronounced.

Figure 13 concurrently modifies the boulders, craters, and roughness intensity on an ideal sphere, showcasing the tool capability in surface features generalization. In each figure of this mosaic, two parameters vary with the values reported in Table 1, while the third morphological feature is kept constant and equal to its minimum value.

While space-based cameras typically operate in grayscale, minor bodies are not necessarily gray. Depending on their spectral type, they can exhibit various color shades. As there is no available information regarding the color of minor celestial bodies, MONET attempts to reproduce the color of some images found online of 67P/Churyumov–Gerasimenko (https://sci.esa.int/web/rosetta/-/55592-hapi-region-on-comet-67p-osiris-nac-false-colour-image, accessed on 14 March 2024), (951) Gaspra (https://www.jpl.nasa.gov/images/pia00125-gaspra-true-and-enhanced-color, accessed on 14 March 2024), (486958) Arrokoth (https://www.nasa.gov/solar-system/far-far-away-in-the-sky-new-horizons-kuiper-belt-flyby-object-officially-named-arrokoth/), accessed on 14 March 2024, and (433) Eros (https://nssdc.gsfc.nasa.gov/planetary/mission/near/near_eros.html, accessed on 14 March 2024). Figure 14 displays the ability of the tool to generate not only highly realistic grayscale images but also colored ones, considering four different shades.

Lastly, MONET can generate new realistic and plausible minor body shapes by interpolating between existing ones and obtaining intermediate shapes. It is important to note that the tool cannot extrapolate new shapes; it can only navigate within the shape space defined by the available three-dimensional models. For representation purposes, in Figure 15, a triangle is considered, with its vertices representing an elongated, flat, and irregular body. The analyzed bodies are 103P/Hartley, (65803) Didymos, and 67P/Churyumov–Gerasimenko, respectively. These models were interpolated with weights equal to 0.25, 0.5, and 0.75. An interpolation with a central sphere was also created with an interpolation weight equal to 0.5. The quantitative metrics defined in Section 2.2 were utilized, revealing a clear trend when transitioning from one body to another. As illustrated in Figure 15b, 103P/Hartley emerges as the most elongated body, with an el parameter of 0.736, gradually decreasing towards (65803) Didymos and 67P/Churyumov–Gerasimenko. In the flatness plot depicted in Figure 15c, the trend for the flatness coefficient fl is not as distinct. All bodies present a similar value of flatness, and, though not in all cases, it changes monotonically. Lastly, as shown in Figure 15d, 67P/Churyumov–Gerasimenko stands out as significantly more irregular, with an irregularity parameter ir of 0.617. Similar to elongation, there is a distinct evolution of irregularity when transitioning between bodies. This quantitative analysis underscores the plausibility of the intermediate models generated with MONET, as their shape values consistently fall within the range defined by the extreme values of the interpolating bodies.

As already specified, the tool allows to generate two configurations of minor body surface features by default, which are the most common characteristics that have been found by visiting these bodies. The two families are distinguished only by a different parameters setting. In detail, the first family, namely rubble-pile, has 300,000 small rocks, 800 medium boulders, and a random number between 1 and 8 for large ones. Two craters types are defined, small and large ones, that are not clearly visible due to the massive presence of rocks that cover them. The second family consists of comet-like surface features; indeed, it is characterized by the alternation of smooth and rough regions, achieved differently from the rocky body thanks to the multiplication of *musgrave* and *noise* textures in the surface roughness generation. These bodies are characterized by the scarce presence of craters, which are also small. The boulders are smaller in size with respect to the other family, and their number is the following: 1000 small rocks, 100 medium boulders, and from 0 to 1, big ones. Some examples of the two families achieved with the tool are shown in Figure 16 and Figure 17, while a summary of the most important parameters characterizing them is given in Table 2.

## 4. Conclusions

This work introduces the design of the DART Minor bOdy geNErator Tool (MONET), an open source Python code (https://github.com/MattiaPugliatti/corto, accessed on 27 March 2024) implemented in Blender for generating realistic minor body models. MONET is used by the DART group for validating and testing IP and VBN algorithms, as well as generating extensive image datasets for artificial intelligence algorithms. For image generation, MONET works in conjunction with CORTO, which handles the positioning of all objects and manages the rendering process. The process of incorporating surface morphology features such as roughness, craters, and boulders is illustrated. MONET enables the creation of a diverse range of morphological features, encompassing both shape and surface characteristics, facilitating the recreation of various minor body conditions. This capability is fundamental given the uncertainty associated with minor bodies. The tool offers the default capability to generate different families of small bodies, namely rubble-pile and comet-like. Additionally, it allows the user to choose morphology parameters that are best suited to their specific application. The authors believe that such tools play a crucial role in evaluating the performance of VBN and IP in ongoing space missions [10,46,47] and projects working on small bodies.

## 5. Future Work

MONET exhibits certain limitations due to its reliance on Blender for modeling physical bodies. One such limitation concerns the size–frequency distribution of boulders on the surface, which is currently managed by the Blender particle system. This system employs three discrete boulder size classes (small, medium, and large) to represent the entire boulder population and randomly places these objects on the body surface. This, together with a poor appearance of a large population of boulders on the surface, fails to accurately reproduce rubble-pile asteroids. For the latter, other tools successfully emulate them, clustering collections of boulders [25]. Additionally, crater modeling relies on textures that do not accurately depict physical crater shapes and freshness. Future enhancements could involve exploring different strategies for the placement and distribution of rocks on the body surface, as well as analytically modeling craters directly on the minor body mesh, thus avoiding the use of textures. Another improvement could entail creating a library of rocks scanned from real ones to increase the fidelity of individual boulders. Moreover, MONET currently lacks optimization for scenarios such as landings and flybys; indeed, the achieved model always represents the highest accuracy attainable. An alternative strategy might involve adjusting the body to reveal more details as the camera approaches it, thereby reducing the overall computational burden. Finally, implementing real-time refinement could allow users to observe changes to the body in real time while adjusting the MONET settings.

## Figures and Tables

**Figure 1 sensors-24-03658-f001:**
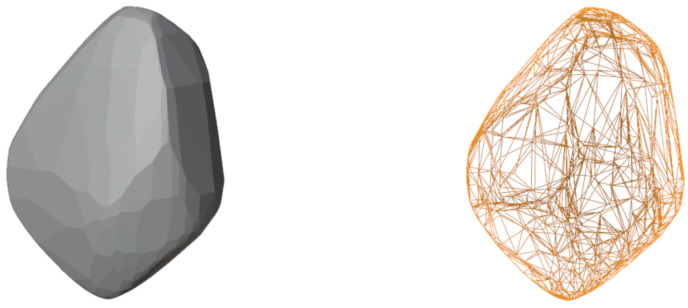
Three-dimensional model visual representation of asteroid (21) Lutetia (**left**) and wireframe representation with mesh artifacts in Blender (**right**).

**Figure 2 sensors-24-03658-f002:**
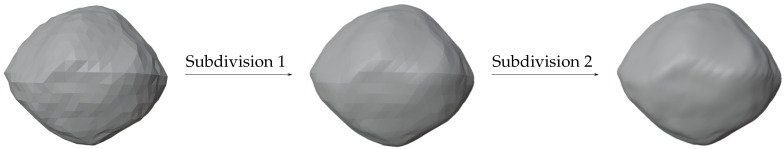
Progressive mesh improvement of the model after subdivision surface modifiers are applied.

**Figure 3 sensors-24-03658-f003:**
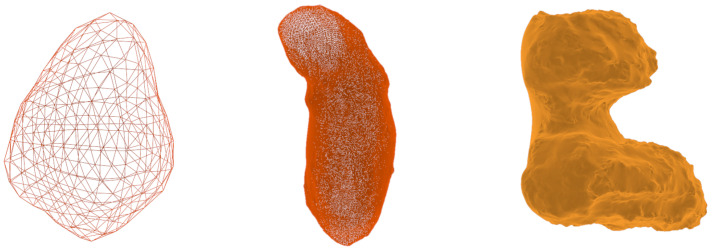
From **left** to **right**, there are minor bodies with an increasing number of faces. On the **left**, asteroid (21) Lutetia (512 faces), in the **middle**, asteroid (25143) Itokawa (49,151 faces) and on the **right**, comet 67P/Churyumov–Gerasimenko (1,309,996 faces).

**Figure 4 sensors-24-03658-f004:**
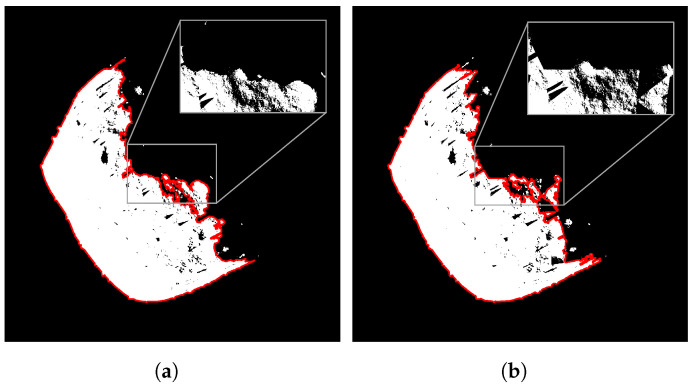
Binarized image of (101955) Bennu asteroid generated with and without the decimate modifier. (**a**) Model with 98,304 faces. (**b**) Model with 768 faces. The red line represents the silhouette of the body.

**Figure 5 sensors-24-03658-f005:**
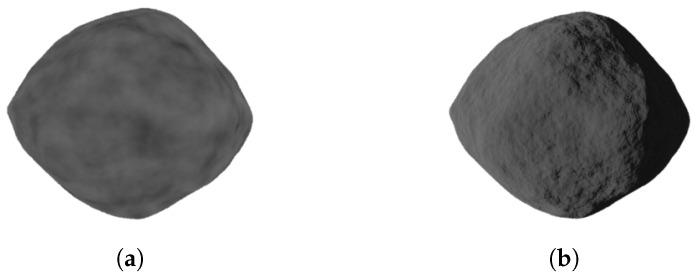
Surface roughness generation. (**a**) Noise texture applied on the model. (**b**) Displacement of the surface exploiting the bump node.

**Figure 6 sensors-24-03658-f006:**
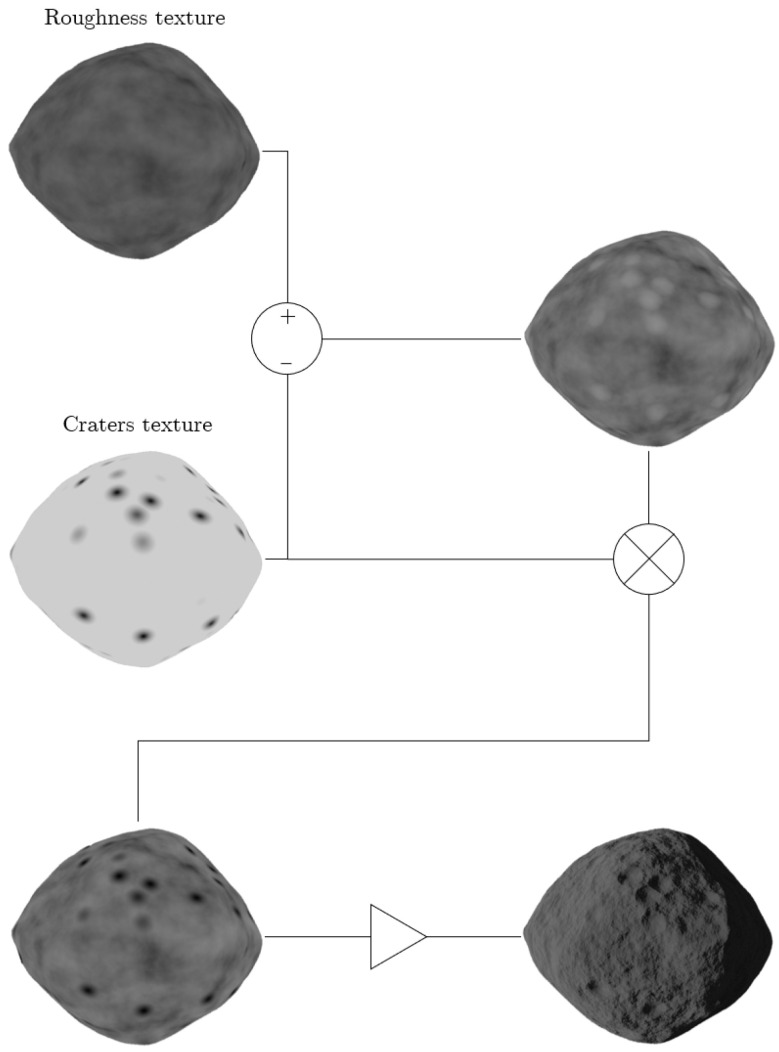
Craters generation procedure. The operations performed on the textures are schematically reported along with the final result depicted on the bottom right. ⊗ represents the mix node multiplication, while ⊳ is the bump node application.

**Figure 7 sensors-24-03658-f007:**
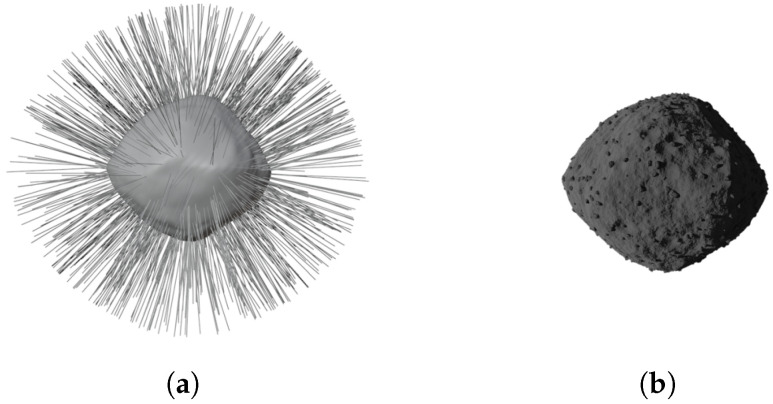
Boulders generation. (**a**) Blender particle system visualization. (**b**) Final result with all the boulders placed on the surface of the model.

**Figure 8 sensors-24-03658-f008:**
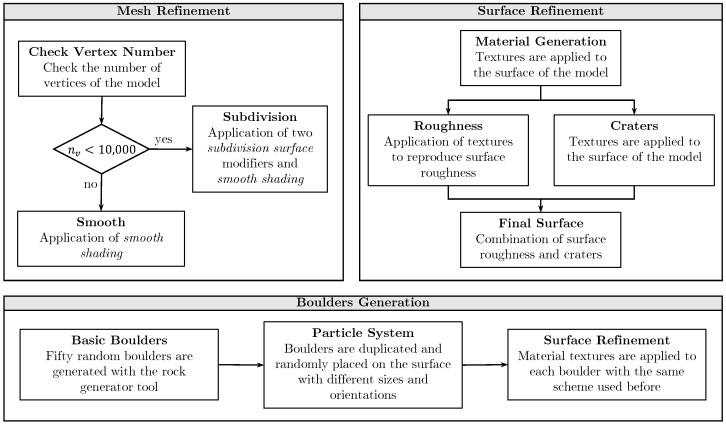
Main building blocks of the Python script to procedurally modify minor bodies surface.

**Figure 9 sensors-24-03658-f009:**
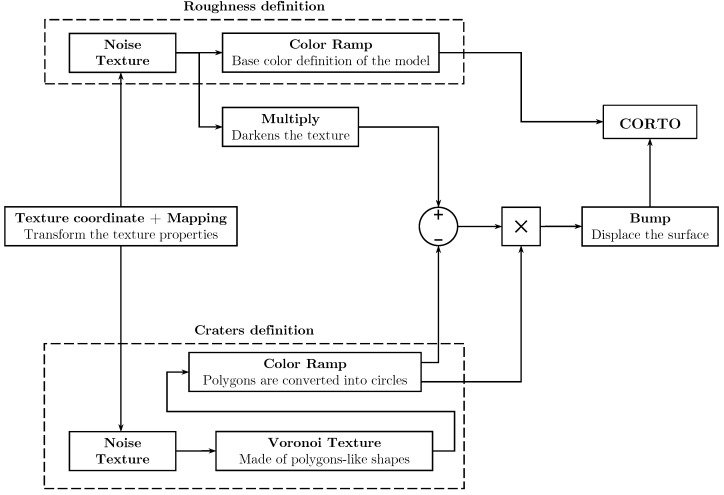
Node tree associated to the surface refinement block for introducing surface roughness and craters.

**Figure 10 sensors-24-03658-f010:**
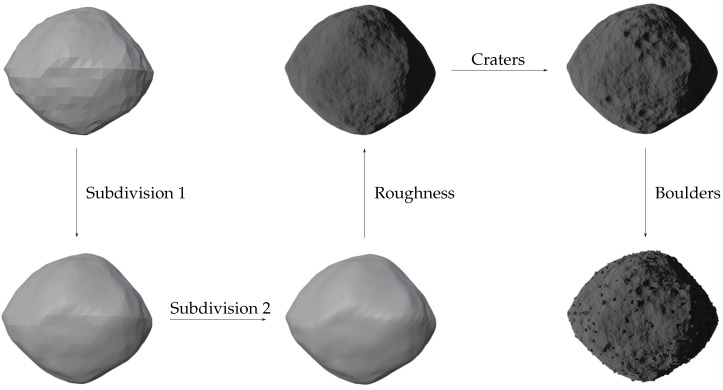
Full process from the starting rough 3D model to the final refined one. First of all, the mesh of the model is improved, exploiting the Blender modifiers. Lately, a rough surface with craters and boulders is obtained.

**Figure 11 sensors-24-03658-f011:**
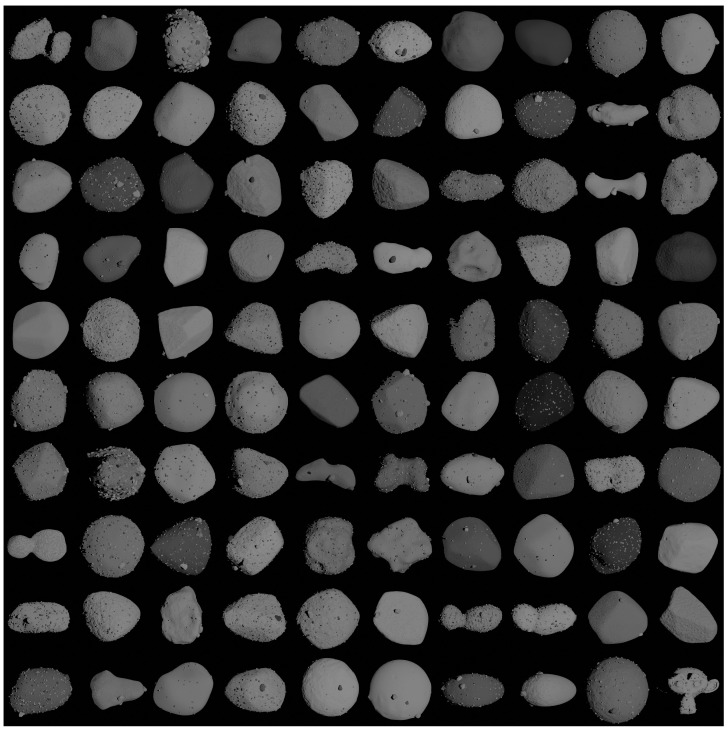
Examples of 100 different augmented shape models generated with MONET.

**Figure 12 sensors-24-03658-f012:**
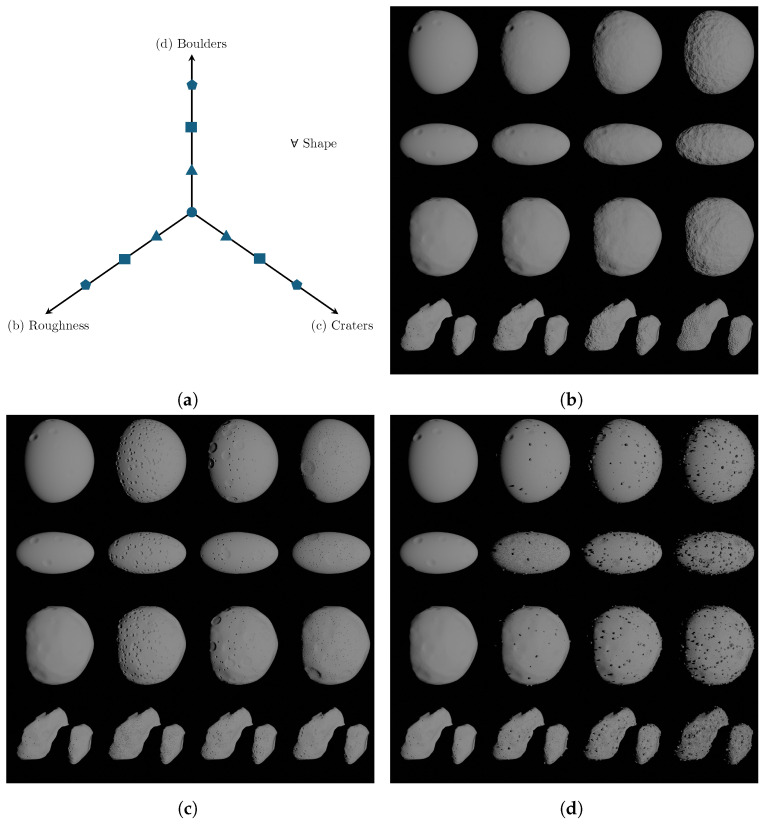
Mosaic 1D: Roughness, craters, and boulders are progressively varied on irregular bodies one at a time. In (**a**), the axes of the schematic represent the configurations of roughness, craters, and boulders explored in the mosaics (**b**–**d**).

**Figure 13 sensors-24-03658-f013:**
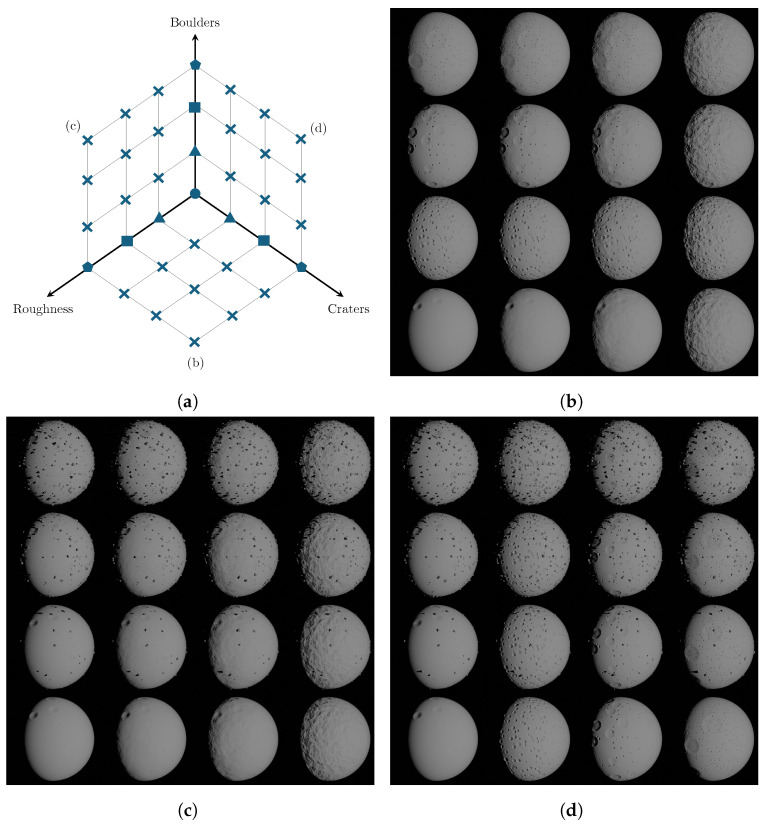
Mosaic 3D: Roughness, craters, and boulders are simultaneously varied on an ideal sphere. In (**a**), the planes of the schematic represent the configurations of roughness, craters, and boulders explored in the mosaics (**b**–**d**).

**Figure 14 sensors-24-03658-f014:**
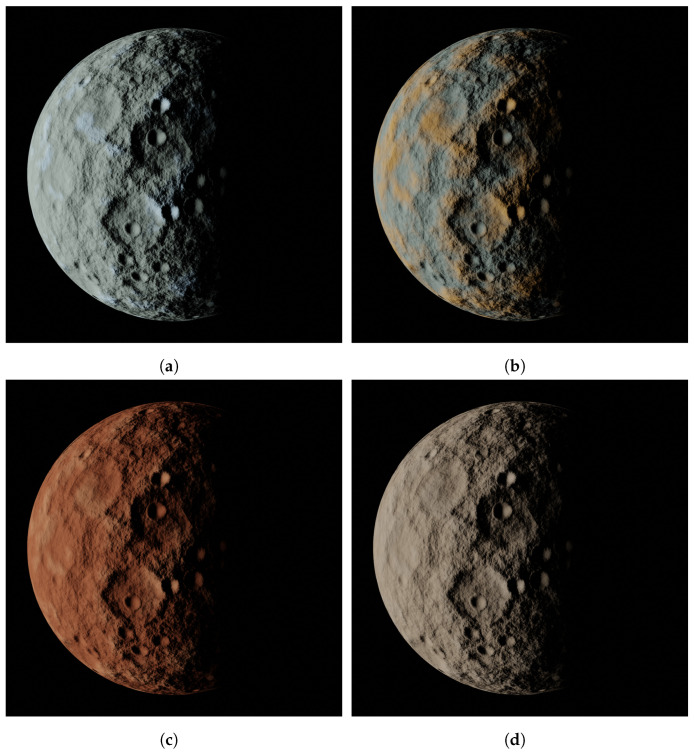
From (**a**–**d**), four different spectral types of asteroid colors that can be achieved in MONET.

**Figure 15 sensors-24-03658-f015:**
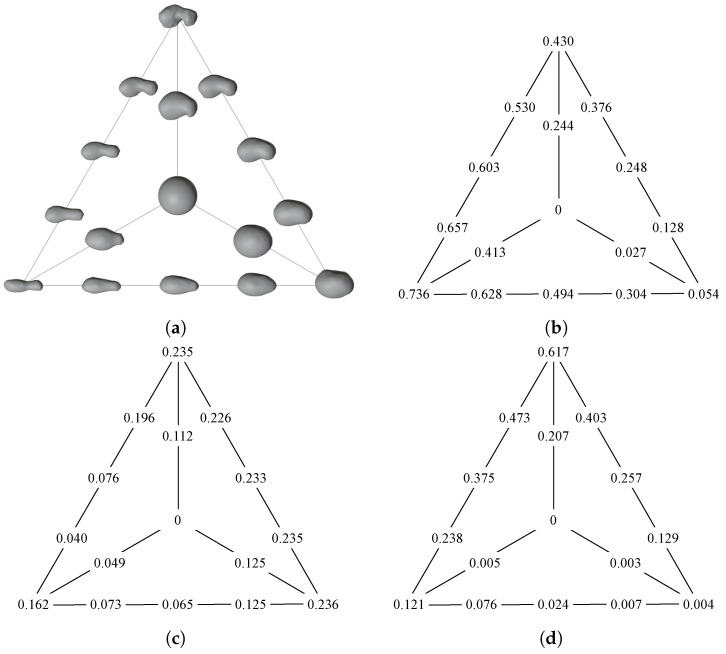
Morphing of four different shapes. In (**a**), the existing minor bodies are located at the edges and the ideal sphere at the center of the triangle of shapes. The lines provide an indication of the morphing direction. The values of the bodies elongation, flatness, and irregularity are shown in (**b**–**d**), respectively.

**Figure 16 sensors-24-03658-f016:**
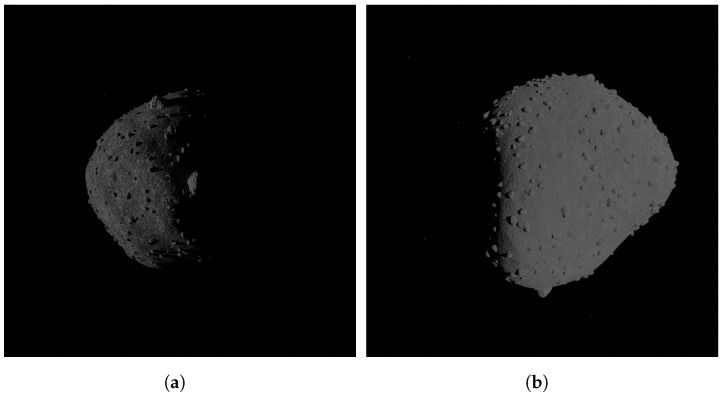
Rendered images with the surface morphological features of the first family. (**a**) (101955) Bennu, (**b**) (88) Thisbe.

**Figure 17 sensors-24-03658-f017:**
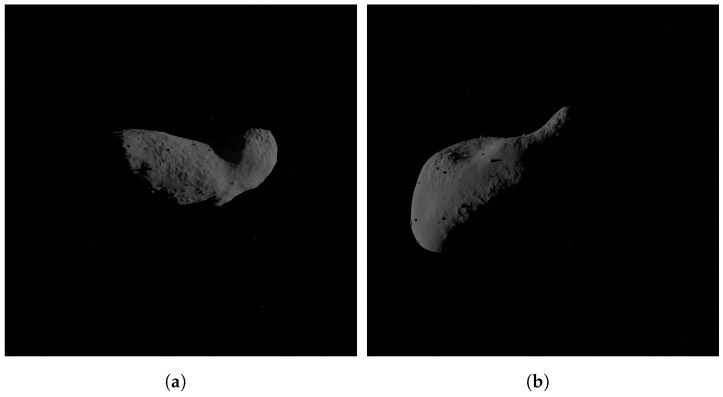
Rendered images with the surface morphological features of the second family. (**a**) (25143) Itokawa, (**b**) (433) Eros.

**Table 1 sensors-24-03658-t001:** Morphological feature values used to build the mosaics.

Feature	Value
	[OΔ□⬠]
Roughness	[0.12515]
Small Craters	[481632]
Large Craters	[01653]
Small Boulders	[01000100,000300,000]
Medium Boulders	[0100300500]
Large Boulders	[0248]

**Table 2 sensors-24-03658-t002:** Minor bodies default families.

		Rubble-Pile	Comet-like
**Roughness**	Texture	Noise	Noise and Musgrave
**Craters**	Dimension	Small and Large	Small
	Texture	Voronoi	Voronoi
**Boulders**	# Small	300,000	1000
	# Medium	800	100
	# Large	1–8	0–1

## Data Availability

Data available on demand.

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
