# Peer review of "MONET: The Minor Body Generator Tool at DART Lab"

_sensors, 2024, doi:10.3390/s24113658_

Round 1

Reviewer 1 Report

Comments and Suggestions for Authors

This paper presents an interesting tool for refining and/or generating synthetic small body geometric models. This is indeed a very important capability for future missions that are increasingly autonomous and rely on ground-validated testing of perception algorithms in photorealistic simulations. The paper is well-written and clearly organized. 

The key missing part of this paper is some sort of validation that the geometries generated are indeed "photorealistic" in some quantitative sense, or at least useful for the purposes of simulation. Consider some metrics by which you can compare your synthetic bodies with real high-resolution shape models. 

As someone who has spent a lot of time looking at real asteroid/comet images (and synthetically generated some myself), I am fairly underwhelmed with the quality of the results shown in the figures. The roughness model, crater shape/freshness model, and boulder morphology and size frequency distribution all seem off. For rubble piles in particular (e.g. Bennu), the surface is generally 100% covered in rocks of different sizes (see DART impact video). MONET should be capable of generating bodies with an apparent 100% rock coverage with an appropriate size-freq. distribution, whether through discrete rock instances, or some roughness/displacement map that simulates rocks. None of the random bodies show are compelling in this regard.

The rock shapes could also be more realistic. Consider building a library of rock assets that are scanned from real rocks.

This tool would also be far more useful if it could scale to "online" and real-time refinement. For example, a simulated close flyby of the surface should be able to reveal detail orders of magnitude smaller than what is visible from an orbital vantage point. 

Author Response

The responses to the reviewers' comments are presented in the attached PDF.

Reviewer 2 Report

Comments and Suggestions for Authors

In the paper “MONET: The Minor bOdy geNErator Tool at DART lab” the Authors describe a method of construction and rendering of minor bodies in space. The developed tool is realized as a Python script for Blender software and provides a possibility to improve the meshing quality and to modify the minor body shapes by means of some hybridization of two bodies. Besides, a considerable attention is paid to creation of realistic surface landscape by implementing roughness, craters and boulders. The purpose of this tool is well explained by the need in testing of Image Processing (IP) and Visual Based Navigation (VBN) software, as well as generation of a large database for the data-driven approaches in this field. In general, the paper is well-written and presents new results within the scopes of Journal. It can be recommended for publication after addressing the following comments:

1. There are a lot of operations with the procedures of the used software “Blander” without a proper description of what they do. See “noise texture is fed into a bump node” as an example. A brief description is required for all similar “operations”, which requires a substantial update of the text. Such explanations will make the text close to a scientific paper, but not a manual.

2. The sentence “Conversely, large craters are initially absent and then their value is decreased” is contradictive. How it can be, value of absent craters is decreased?

3. Fig. 11: the bottom right model seems strange. What is it?

4. Fig. 12: boulders seem unrealistic, because they touch the surface rather than being pressed into it like real boulders.

5. Text in Figures 18-21 is unreadable; therefore, they are useless in the present form.

Author Response

(The authors gave the same response as above.)

Round 2

Reviewer 1 Report

Comments and Suggestions for Authors

Thank you for addressing all my comments. The additions and clarifications made within the paper help to clarify what the tool can and cannot do. However, since all of my recommendations have been pushed to future work, it is not clear that this contribution is a substantive improvement in the state of the art. The general process of morphology + craters + roughness + boulders is a standard technique and the Blender implementation is not particularly novel and has has substantial limitations. Without any validation of these shape models (i.e. running them through representative autonomy algorithms and showing they have similar features to real small bodies), it is also unclear how useful the tool is in its current form. 

Author Response

The authors acknowledge that the combination of morphology, craters, roughness, and boulders is a standard technique, and the implementation in Blender is not novel and has significant limitations. However, as explained in the introduction, many related works and tools are either not open source or have limitations addressing only a few morphological features or shapes, rather than all at once. Additionally, while we are aware of Blender's limitations, it was chosen for three main reasons: it is open source, has a user-friendly interface, and has a strong community.

We want to emphasize that the primary purpose of MONET is not to precisely replicate the morphology of existing minor bodies. Instead, its final aim is to allow the testing of VBN and IP algorithms under challenging morphological conditions and shapes, which may not necessarily represent real physics. This approach allows us to assess the robustness of algorithms in unexpected scenarios and understand their capabilities in handling vastly different conditions. It's worth mentioning that the code will be open source, enabling the community to understand and modify it according to their needs. For instance, if a specific distribution of boulders on the surface is required, users can write their own code and integrate it into the boulder generation function.